# Cytotoxic Isopentenyl Phloroglucinol Compounds from *Garcinia xanthochymus* Using LC-MS-Based Metabolomics

**DOI:** 10.3390/metabo13020258

**Published:** 2023-02-10

**Authors:** Fan Quan, Xinbo Luan, Jian Zhang, Wenjie Gao, Jian Yan, Ping Li

**Affiliations:** Key Laboratory of Agro-Environment in the Tropics, Ministry of Agriculture and Rural Affairs, Guangdong Provincial Key Laboratory of Eco-Circular Agriculture, Guangdong Engineering Research Centre for Modern Eco-Agriculture, College of Natural Resources and Environment, South China Agricultural University, Guangzhou 510642, China

**Keywords:** *Garcinia xanthochymus*, metabolomics, cytotoxicity, biomarker, LC-MS, isopentenyl phloroglucinols

## Abstract

Many unique chemical metabolites with significant antitumor activities have been isolated from *Garcinia* species and have become a leading hotspot of antitumor research in recent years. The aim of this study was to identify bioactive compounds from different plant parts (leaf, branch, stem bark, fruit, and seed) of *G. xanthochymus* through combining LC-MS-based metabolomics with cytotoxicity assays. As a result, 70% methanol seed extract exerted significant cytotoxic effects on five human cancer cell types (HL-60, A549, SMMC-7721, MDA-MB-231, and SW480). LC-MS-based metabolomics analysis was used, including principal component analysis (PCA) and orthogonal partial least squares discriminant analysis (OPLS-DA), in order to identify 12 potential markers from seed extract that may relate to bioactivity. LC-MS guidance isolated the markers to obtain three compounds and identified new isopentenyl phloroglucinols (**1**–**3**, named garxanthochin A–C), using spectroscopic methods. Among them, garxanthochin B (**2**) demonstrated moderate inhibitory activities against five human cancer cell types, with IC_50_ values of 14.71~24.43 μM. These findings indicate that *G. xanthochymus* seed has significant cytotoxic activity against cancer cells and garxanthochin B has potential applications in the development of antitumor-led natural compounds.

## 1. Introduction

Tumors are one of the multiple diseases that seriously threaten human health. The GLOBOCAN 2020 estimate showed that in 2020, there were 19.3 million new cancer cases, and nearly 10 million people died of cancer. Assuming that the 2020 estimated national incidence rate remains unchanged, it is estimated that 28.4 million new cancer cases will appear in the world in 2040, an increase of 47% over the corresponding 19.3 million cases in 2020 [1]. Regardless of the level of human development, this disease is an important cause of the incidence rate and mortality around the world [1,2]. Currently, the main treatment for cancer includes radiotherapy, chemotherapy, and surgery [3]. Studies have shown that natural products have rich antitumor active components and are an important source of antitumor drugs, which may be related to their rich and diverse chemical types and pharmacological activities [4]. Among the 1881 new drugs that have been approved by the Food and Drug Administration (FDA) over the past 40 years, 23.5% were natural product molecules. It can be seen that natural products play an important role in the field of new drug research and development [5]. Nevertheless, the traditional isolation and characterization procedures for natural products are challenging to perform due to their complexity and the limited amounts of metabolites in the extracts [6,7]. Therefore, targeted isolation is used to reduce the drawbacks in natural-product-based drug discoveries in recent years. Metabolomics strategies, assisted by sensitive analytical techniques and multivariate analyses, seem to be an efficient approach to discover the biomarkers of natural products [8]. LC-MS-based metabolomics, combined with metabolomics analysis including unsupervised methods (PCA) and supervised methods (OPLS-DA), have successfully been applied for identifying potential bioactive natural products [9,10].

*Garcinia xanthochymus*, also called gamboge, yellow mangosteen, and false mangosteen, is a medicinal and edible plant that belongs to the Clusiaceae family [11]. It is distributed in Polynesia, Southeast Asia, Africa, Australia, North Thailand, Myanmar, and Yunnan in China. It grows as a tropical tree species whose fresh mature fruits can be eaten directly or used for making jam, vinegar, beverages, and other products. It is also used to make watercolor paintings and yellow dyes for fabrics [12]. Traditionally, *G. xanthochymus* is widely used to treat diarrhea, dysentery, nausea, and vomiting [13]. In China, it is a traditional Dai medicine in Yunnan province and is commonly used to dispel worms and eliminate food toxins [14]. The fruit is rich in nutrients such as carbohydrates, protein, fat, vitamins, and minerals, as well as polyisoprenylated benzophenones, phenol, and flavonoids [13,15,16]. Many beneficial phytochemical components are found in the stem bark, leaves, branches, and seeds of this plant [17]. Phytochemical studies have demonstrated that a number of chemical constituents, including biflavonoids, xanthones, benzophenones, and phloroglucinols, display a range of pharmacological activities, such as antitumor, cytotoxic, antioxidant, anti-inflammatory, and antidiabetic activities [15,18]. Our previous study showed that the metabolites from *G. xanthochymus* fruit displayed antiangiogenic activity [19], which may have potential in combatting tumor development. These findings led to our further investigations on the cytotoxic activities of metabolites from different parts of *G. xanchochymus*.

In this study, different parts of *G. xanthochymus* were assessed for their cytotoxic activities against five human cancer cell lines, including HL-60 (leukemia), SMMC-7721 (hepatoma), A-549 (lung cancer), MCF-7 (breast cancer), and SW480 (colon cancer). Meanwhile, the chemical constituents of different parts of *G. xanthochymus* were analyzed using ultra-high-performance liquid chromatography (UPLC) coupled to quadrupole time-of-flight mass spectrometry (QTOF-MS). Metabolomics analysis, integrated with bioassay results to guide isolation, resulted in three new compounds from *G. xanthochymus* seeds, which exhibited significant cytotoxic activity. Our study helps to further consider the utilization of bioactive metabolites of *G. xanthochymus.*

## 2. Materials and Methods

### 2.1. General Experimental Procedures

NMR spectra at 600 MHz for ^1^H and 151 MHz for ^13^C were obtained using a Bruker AVANCE-600 (600 MHz) instrument (Bruker Biospin, Zurich, Switzerland). HR-ESI-MS spectra were obtained from UPLC-QTOF-MS (Xevo G2 QTOF, Waters MS Technologies, Manchester, UK). Semipreparative HPLC was conducted on an Agilent 1100 HPLC, equipped with a Zorbax 300 SB-C18 column (9.4 mm × 25 cm, 4 μm). CD spectra were detected from a Chriascan spectrometer (Applied Photophysics Ltd., Leatherhead, UK). The absorbance was detected on an Multiskan FC microplate reader (Thermo Fisher, US.). Column chromatography was performed over Sephadex LH-20 (GE Healthcare Bio-Sciences AB, Sweden), thin-layer chromatography (TLC), and silica gel (300–400 mesh, Qingdao Haiyang Chemical Co. Ltd. (Qingdao, China)). Chromatographic-grade acetonitrile, methanol, and formic acid were purchased from J. T. Baker (Phillipsburg, NJ, USA). The reagents used in bioactivity assays including corticosterone Dulbecco’s modified Eagle’s medium (DMEM), dimethyl sulfoxide (DMSO) from Sigma-Aldrich, 3-(4,5-dimethylthiazol-2-yl)-5(3-carboxymethoxyphenyl)2-(4-sulfopheny)-2H-tetrazolium (MTS) from Promega (Madison, WI, USA), and PBS (phosphatic buffer solution) from Biological Industries.

### 2.2. Plant Materials

The different parts of *G. xanthochymus* including leaves, branches, stem bark, fruit, and seeds were collected in Xishuangbanna, Yunnan province, China, and voucher specimens were verified by Dr. Tieyao Tu (South China National Botanical Garden). The voucher specimens were deposited at the South China Agricultural University Herbarium. Other collected samples were stored at −40 °C until extraction.

### 2.3. Extraction and Isolation

The leaf, branch, stem bark, and seed were naturally air-dried, and the fruit was freeze-dried. All dried samples were grounded into a powder and extracted with 70% methanol followed by ultrasonic extraction at room temperature for 30 min. Each sample was extracted three times with the same method; the filtrate was concentrated in a vacuum with a rotary evaporator to obtain dry residues of different parts that were then stored at 4 °C until their use. The extracts were dissolved with chromatographic-grade methanol to obtain a concentration of 1 mg/mL of crude extract, and were filtered through a 0.22 µm PTFE syringe filter and transferred into a 2 mL autosampler vial for UPLC-QTOF-MS analysis. The quality control (QC) sample was prepared by mixing 50 μL of each methanol solution with different part extracts.

Chemical constituents were isolated from the bioactive extracts of seeds as follows: the 70% methanol of the seed extract was suspended in pure water and then extracted with petroleum ether (PE) for three times. The filtrate was combined and concentrated to obtain 11 g of the PE extract. The extract (1 g) was subsequently purified via semipreparative HPLC, and gradient-eluted with MeCN/H_2_O (3 mL/min) to provide compound **1** (13 mg). At the same time, the PE extract (10 g) was fractionated with silica gel column chromatography, and 18 fractions (s-pet-1~s-pet-18) were obtained using gradient elution with petroleum ether acetone (100:0 → 0:100). Fraction s-pet-3 (1 g) was separated with Sephadex LH-20 (dichloromethane-methanol, *v/v* = 1:1) to obtain 8 fractions (s-pet-3-1~s-pet-3-8). S-pet-3-5 was purified using semipreparative HPLC and gradient-eluted MeCN/H_2_O to obtain compound **2** (7.8 mg). Fraction s-pet-4 (540 mg) was separated with Sephadex LH-20 (dichloromethane-methanol, *v/v* = 1:1) to obtain 10 fractions (s-pet-4-1~s-pet-4-10). S-pet-4-7 (98 mg) was separated by semipreparative HPLC and gradient-eluted with MeCN/H_2_O to obtain 9 fractions (s-pet-4-7-1 ~ s-pet-4-7-9), and s-pet-4-7-4 (12 mg) was purified using analytical column HPLC and gradient-eluted with MeCN/H_2_O to obtain compound **3** (4 mg). All the solutions of target peaks in semipreparative HPLC were thoroughly dried using nitrogen.

### 2.4. Cell Viability Assay

Human cancer cell lines including HL-60 (peripheral blood), A-549 (lung), MCF-7 (breast cancer), and SW480 (colon cancer) were obtained from ATCC (Manassas, VA, USA); SMMC-7721 (hepatoma) was purchased from BeNa Culture Collection (BNCC, Beijing, China). The cells were cultured in RMPI-1640 or DMEM medium (Biological Industries, Kibbutz Beit-Haemek, Israel) and supplemented with 10% fetal bovine serum (Biological Industries) at 37 °C in a humidified atmosphere with 5% CO_2_. The cytotoxicity assay was evaluated via an MTS assay [20]. Briefly, 100 μL of adherent cells was seeded into each well of a 96-well cell culture plate. After 12 h of incubation at 37 °C, the crude extract (100 μg/mL) or test compound (40 μM) was added with an initial density of 1 × 10^5^ cells/mL in 100 μL of the medium. The experiments were performed in triplicate, with cisplatin and paclitaxel (Sigma) as positive controls. Test samples with a growth inhibition rate of 50% were further evaluated at concentrations of 100 μg/mL, 20 μg/mL, 4 μg/mL, 0.8 μg/mL, and 0.16 μg/mL, in triplicate. After being incubated for 48 h, the MTS solution (20 μL) and culture medium (100 μL) were added into each well to continue incubating for 4 h. The optical densities of test samples were measured at a wavelength of 490 nm using a Multiskan FC Microplate Photometer (Thermo Scientifc). The IC_50_ value of each compound was calculated with Reed and Muench’s method. The results were analyzed using GraphPad Prism (version 8.0, La Jolla, CA, USA).

### 2.5. UPLC-QTof-MS Analysis

Metabolite analyses of different plant parts were performed on an ACQUITY UPLC system coupled with a Xevo G2-XS QTof spectrometer (Waters Milford, MA, USA). An aliquot of 0.5 μL was injected into an ACQUITY UPLC^®^ BEH C_18_ column (2.1 mm × 50 mm, 1.70 μm; Waters), and the column temperature was kept at 40 °C. The mobile phase solvents consisted of water with 0.1% formic acid (A) and acetonitrile with 0.1% formic acid (B). The linear gradient elution was performed as follows: 0–1.0 min, 15–20% B; 1.0–3.0 min, 20–60% B; 3.0–5.0 min, 60–75% B; 5.0–7.5 min, 75–80% B; 7.5–8.5 min, 80–95% B; 8.5–10.0 min, 95% B; 10.0–10.1 min, 95–15% B; and 10.1–12 min, 15% B. The flow rate was set at 0.3 mL/min. The autosampler was conditioned at 16 °C.

High-resolution mass spectra were recorded with a Xevo G2-XS QTof-MS system coupled with an electrospray ionization source, and the analyzer was set to sensitivity mode. The capillary voltage was set to either +3.0 kV and −1.50 kV; the source temperature and desolvation temperature were maintained at 120 °C and 400 °C, respectively. The cone gas and desolvation gas flow were 50 L/h and 800 L/h, respectively. Mass data were collected using MS^E^ in positive and negative modes using a scan time of 0.5 s and a range of *m*/*z* 100–1000 Da. The low energy was set as 5 V to obtain the precursor ion spectrum and a ramp collision energy from 20 V to 50 V was set to acquire fragmentation data in a single analysis. Leucine encephalin was applied as the lock mass.

### 2.6. Metabolomics Data Analysis

The negative data obtained from the UPLC-QTOF-MS analysis of different parts from *G. xanthochymus* were analyzed with MS-DIAL 4.60 software [21]. The raw data had to be converted to ABF format with an ABF converter for data processing. MS-DIAL software was used, and the main parameter settings were as follows: a retention time range of 0.3–10 min; a mass range of 100–1000 Da; the minimum peak height was set at 10,000 amplitudes; and the sigma window value was 0.5. The QC sample was used as an alignment reference file. The generated metabolite output data matrix, with the corresponding RT, *m/z*, and peak area acquired for each sample, was subjected to further metabolomics analysis. The output data from MS-DIAL were imported into the SIMCA software (version 14.1, Umetrics, Umea, Sweden). Untargeted principal component analysis (PCA) and supervised orthogonal partial least squares discriminate analysis (OPLS-DA) were performed with Pareto scaling. Hotelling’s 95% was shown as an ellipse in the score plots; the different metabolites were identified via an S alphabet-like plot (S-plot), which was calculated to visualize the relationship between covariance and correlation among the OPLS-DA results. A heatmap analysis was performed using MetaboAnalyst 5.0 https://www.metaboanalyst.ca/ accessed on 27 May 2021.

## 3. Results and Discussion

### 3.1. Cytotoxic Activity Screening of Different Part Extracts of G. xanthochymus

Five human cancer cell lines (HL-60, A549, SMMC-7721, MCF-7, and SW480) were cultured with different parts (leaf, branch, stem bark, fruit, and seed) of *G. xanthochymus* crude methanol extracts at 100 μg/mL. MTS assays were performed to measure the cell viability in the culture medium. The proportions of 70% methanol extract of different parts were evaluated for their cell inhibition against five human cancer cell types (Figure 1). The seed extract had the most significant cytotoxic effects on five cancer cell types. In contrast, the leaf, branch, and fruit extracts had no significant effect on the cell viability. These cytotoxicity studies demonstrated that the seed methanol extracts were the most active. Moreover, the cytotoxicity of these extracts against five human cancer cell lines is shown in Table 1. The 70% methanol seed extract had a good cytotoxic effect on HL-60, A549, SMMC-7721, MDA-MA-231, and SW480, with IC_50_ values of 8.637 ± 0.124, 3.436 ± 0.169, 2.666 ± 0.130, 8.362 ± 0.562, and 9.438 ± 0.290 μg/mL, respectively. These results demonstrated that the cytotoxic activity of the 70% methanol seed extract from *G. xanthochymus* underwent reasonable bioactivity-guided fractionation to identify the cytotoxic activity compounds.

### 3.2. Metabolites in Different Parts of G. xanthochymus and Marker Compounds’ Identification

The preliminary activity results suggested that the 70% methanol extract of *G. xanthochymus* seed exhibited the most cytotoxic activities, while the extracts of the other parts extracts did not show promising inhibitory effects. To rapidly identify marker compounds from the seed extract relative to their bioactive substances, the metabolic profiling of different parts of *G. xanthochymus* was carried out through UPLC-QTOF-MS in both positive and negative modes. By optimizing the UPLC-QTOF-MS conditions, the optimal test methods were provided for different parts of *G. xanthochymus*, such as the elution gradient and the concentration of formic acid in the mobile phase, in order to obtain the best chromatographic peak intension and separation. The negative mode was selected for analysis because more peaks were detected, it had more sensitivity, there was less background noise, and their relative abundance was higher compared to the positive mode. Ultimately, the representative chromatograms of the leaf, branch, stem bark, fruit, and seed are shown in Figure 2.

The retention time and negative molecular ion peaks of the main components in the extracts of different parts were obtained through UPLC-QTOF-MS analysis. In addition, the high-resolution mass spectra of molecular ions and fragment ions of metabolites were obtained from the low energy and ramp collision energy of MS^E^ data. Based on ion fragmentation rules, comparisons with open source mass spectrometry databases, and a literature comparison with *Garcinia* species, 47 compounds including 14 biflavonoids, 6 xanthones, 10 phloroglucinols, and others were tentatively identified, as shown in Appendix A.

The heatmap representation was used on the basis of the peak areas of the identified metabolites. There were significant differences in the expressions of metabolites between different parts of *G. xanthochymus* (Figure 3). All metabolites were divided into two categories; the red type indicated that the metabolites were relatively high in content. The distribution of metabolites in different parts of *G. xanthochymus* was generally partitioned into four groups: phloroglucinols, xanthones, flavonoids, and biflavonoids. Phloroglucinols were present in the highest levels in the seed, with low content in the branch, leaf, stem bark, and fruit. For example, the compounds garcinielliptone F, garcinielliptone T, and garsubellin B were present in significant quantities in the seed. Meanwhile, xanthones were mainly present in the stem bark and branch, but the content was lower in other parts. Biflavonoids and flavonoids were present in higher levels in the leaf, but the expression levels in other parts were relatively low. The study on the metabolic profiles of *G. xanthochymus* enabled us to discover that the seeds were rich in phloroglucinol compounds, most of which were reported in this plant for the first time. In general, based on the heatmap analysis, we could intuitively observe different kinds of chemical components and their content levels in the branch, leaf, stem bark, seed, and fruit of the plant.

The potential active markers of methanol extracts from different parts of *G. xanthochymus* were identified via UPLC-QTOF-MS combined with metabolomics analysis. An unsupervised statistics PCA was used to assess whether any differences in the metabolomes of the different parts were detectable [22]. In the PCA score plots, each point represents a single sample. Samples with similar chemical composition are clustered together, while samples with different chemical composition are scattered [23]. In order to study the metabolite composition of the branch, leaf, fruit, stem bark, and seed of *G. xanthochymus,* we conducted an untargeted metabolite profiling approach using UPLC-QTOF-MS, with total ion chromatograms (TIC). As shown in Figure 4A, three replicate sample data (negative) from different parts were clustered together, and there was no overlap of active and inactive clusters from the branch, leaf, fruit, stem bark, and seed. Through the biological activity screening results, with seeds as the active group and other parts as the inactive group, the S-plot of OPLS-DA highlighted the ions that caused the active and inactive plant parts, with each point representing a retention time and exact mass pair (Figure 4B). The farther these data points were from the origin, the greater the contribution to the difference between these groups (active and inactive). The upper-right quadrant of the S-plot played a great role in distinguishing active extracts and inactive extracts against cancer cells. Based on the metabolomics analysis, the fragmentation rules, the comparison of open source mass spectrometry databases, and the literature comparison with *Garcinia* species, 12 potential markers that may be related to the activity of these extracts were identified, and these are shown in Table 2. These compounds included one friedolanostane, eight phloroglucinols, and three unknown compounds.

### 3.3. Isolation and Structure Elucidation of Metabolites from Target Markers

We used bioassay-guided and LC-MS-guided results to try to isolate 12 markers (Table 2). As a result, three compounds were isolated from *G. xanthochymus* seed, with structure identification via NMR, mass spectrometry, and electrostatic circular dichroism (ECD). All of the isolated compounds were new polycyclic isopentenyl phloroglucinols, named garxanthochin A (**1**), garxanthochin B (**2**), and garxanthochin C (**3**), respectively. The structures of compounds **1**–**3** are shown in Figure 5. The identification process used to obtain the new compounds is described below.

Compound **1** was obtained as a yellow oil. The high-resolution electroscope ionization mass spectra (HR-ESI-MS) showed a pseudomolecular ion peak at *m/z* 483.3121 [M+H]^+^ (calculated C_30_H_43_O_5_, *m/z* 483.3110 [M+H]^+^), corresponding to the molecular formula C_30_H_42_O_5_ and 10 degrees of unsaturation. The analysis of ^1^H NMR, ^13^C NMR, and DEPT data (Table 3) showed that **1** contained a group of typical signals of phloroglucinol [δ_C_ 186.9 (C-1), δ_C_ 106.8 (C-2), δ_C_ 171.4 (C-3), δ_C_ 57.8 (C-4), δ_C_ 196.4 (C-5), δ_C_ 107.3 (C-6)]. Inspection of the ^1^H-^1^H COSY and HMBC spectra led to the identification of an isobutyryl (C-27 to C-30) and a geranyl unit (C-12 to C-21). The HMBC correlations of Me-25 and Me-26 with the oxygenated C-24, H-23 with C-25 and C-24 defined an oxygenated prenyl unit (C-22 to C-36). The HMBC correlations of H-7/C-4, C-3, C-9 and C-8, H-10/C-11, C-9 and C-8, H-11/C-10, C-9 and C-8, proved that the fragment was an isopentenyl unit. In addition, a set of key HMBC correlations (H-7 with C-4, C-3) and the deshielded shifts of C-3 (δ C171.4) and C-10 (δ C61.8) suggested an *O*-bridge between C-3 and C-10 to fulfill the degree of unsaturation, which proves that **1** had a seven-membered ring, and that the seven-membered ring was connected to the six-membered ring through C-4 and C-3. As a result of the HMBC correlation between H-12 and C-5, C-4 and C-7, it was proved that the two connected isopentenyl units were connected to C-4. The HMBC correlation between H-22 and C-1, C-3, H-23, and C-2 proves that the oxygenated prenyl unit was connected to C-2. The remaining isobutyryl unit was connected to C-6. Through a Scifinder database comparison, **1** was identified as a new polycyclic isopentenyl phloroglucinol and named garxanthochin A. Its key ^1^H-^1^H COSY and HMBC correlations are shown in Figure 6.

Compound **2** was obtained as a yellow oil. High-resolution electrospray ionization mass spectra (HR-ESI-MS) showed a pseudo-molecular ion peak at *m/z* 497.3270 [M+H]^+^ (calculated C_31_H_45_O_5_, *m/z* 497.3267 [M+H]^+^), corresponding to the molecular formula C_31_H_44_O_5_ and 10 degrees of unsaturation. Analysis of ^1^H NMR, ^13^C NMR and DEPT data (Table 3) showed that **2** had 11 quaternary carbons and 20 protonated (eight methyl, six methylenes, and six methines) carbon atoms. Compound **2** contained a group of typical signals of phloroglucinol [δ_C_ 187.0 (C-1), δ_C_ 106.8 (C-2), δ_C_ 171.4 (C-3), δ_C_ 57.8 (C-4), δ_C_ 196.4 (C-5), δ_C_ 107.3 (C-6)]. However, in comparison to **1**, the molecular formula of **2** was found to contain 14 more mass units, suggesting the presence of an extra methylene. The available NMR data of **1** were compared with those derived from the ^1^H NMR, ^13^C NMR and HMBC spectra of **2**; it was found that both compounds **1** and **2** had the same parent nucleus. As expected, compound **2** also had a seven-membered ring connected to the six-membered ring, and one geranyl group along with a oxygenated prenyl unit was determined. ^1^H-NMR data showed that δ_H_ 1.35 (H-30) of **2** was different from **1** at δ_H_ 1.12 (H-30), and in the ^1^H-^1^H COSY spectrum of **2**, H-30 (δ_H_ 1.35) and H-28 (δ_H_ 3.81) had a relevant signal. In the HMBC spectrum, H-30 (δ_H_ 1.35) was correlated with C-28 (δ_C_ 42.1), C-29 (δ_C_ 16.4), C-27 (δ_C_ 207.6), and C-31 (δ_C_ 12.0), and H-31 (δ_H_ 0.89) was related to C-28 (δ_C_ 42.1) and C-30 (δ_C_ 26.9), indicating that C-30 was connected to a methyl group. Its key ^1^H-^1^H COSY and HMBC correlations are shown in Figure 6. In conclusion, the structure of **2** was determined and identified as a new polycyclic isopentenyl phloroglucinol and named garxanthochin B.

Compounds **1** and **2** have relatively novel structures and a parent nucleus composed of a six-membered ring and a seven-membered heteroxygen ring. The C-4 of their planar structures has chiral configuration. The absolute configurations of **1** and **2** were determined by comparing the experimental measured diagram of the CD spectrum and the calculated simulation diagram of ECD. The calculated curve for the *R* stereoisomer paralleled the experimental spectrum of **1**, thus defining the absolute configuration of the molecule (Figure 7). In addition, the structure and experimental CD spectra of compounds **1** and **2** were basically similar. Thus, the absolute configuration of **2** was also an *R* stereoisomer.

Compound **3** was obtained as a yellow oil. High-resolution electrospray ionization mass spectra (HR-ESI-MS) showed a pseudomolecular ion peak at *m/z* 499.3077 [M-H]^−^ (calculated C_30_H_43_O_6_ *m/z* 499.3060 [M+H]^+^), corresponding to the molecular formula C_30_H_44_O_6_, and nine degrees of unsaturation. Analysis of ^1^H NMR, ^13^C NMR, and DEPT data (Table 3) showed that **3** had 11 quaternary carbons and 19 protonated (eight methyl, five methylenes, and six methines). Compared with the remaining NMR data of 4-hydroxycolupulone [31], it was found that compound **3** and 4-hydroxycolupulone have the same six-membered ring parent nucleus structure and an isobutyryl unit (C-27 to C-30) and an isopentenyl unit (C-22 to C-26). Inspection of the ^1^H-^1^H COSY and HMBC spectra led to the identification of a geranyl unit (C-7 to C-16), which was also supported by the HMBC correlation of H-11 with C-10, C-9, and H-10 with C-11 and C-12. The HMBC correlations of H-20, H-21, and H-17 with both C-18 and C-19, and H-18 with both C-20 and C-21, defined an oxygenated prenyl unit. Due to the HMBC correlation of H-7 with C-3, C-1, C-2, and C-17, and H-17 with C-3, C-1, C-2, and C-7, it was proved that the oxygenated prenyl unit and geranyl group were connected in C-2. The HMBC correlations of H-22 with C-1, C-5, and C-6 demonstrated that a single isopentenyl unit was attached at C-6. The remaining isobutyryl unit was attached at C-4. Finally, compound **3** was identified as a new polycyclic isopentenyl phloroglucinols and named garxanthochin C. Its key ^1^H-^1^H COSY and HMBC correlations are shown in Figure 6.

### 3.4. Cytotoxic Effect of Compounds Isolated from G. xanthochymus Seed

Based on the markers analyzed via the OPLS-DA model, three markers (compounds **1**–**3**) were obtained through targeted separation. The cytotoxicity assay was performed on the isolated markers **1**–**3** in order to verify whether the markers analyzed by the model were related to the activity. The cytotoxic effect on five human cancer cell lines (HL-60, A549, SMMC-7721, MDA-MB-231, and SW480) of three markers **1**–**3** was assessed, as shown in Figure 8. Compound **2** exhibited strong cytotoxic activity against five human cancer cell types, with inhibition rates of 83.56%, 86.33%, 91.14%, 98.70%, and 88.47% under a treatment concentration of 40 μM, respectively. Notably, compound **2** showed the strongest inhibitory activity on MDA-MB-231 cells. Compound **1** also possessed moderate inhibitory activities for A549, SMMC-7721, MDA-MB-231, and SW480 cells, with cell inhibition rates of 44.79%, 37.83%, 41.30%, and 37.38%, respectively. Compound **3** showed weak inhibitory activity for A549 and SMMC-7721, with cell inhibition rates of 29.00% and 29.09%, respectively. Therefore, compound **2** was selected for further bioassay screening, and showed significant cytotoxicity against five human cancer cell lines (Table 4), with IC_50_ values of 19.02 ± 1.26, 24.43 ± 0.42, 15.69 ± 0.57, 16.29 ± 1.44, and 14.71 ± 0.33 μM, respectively. The inhibitory effect was similar to that of the positive control cisplatin.

## 4. Discussion

The *Garcinia* genus has high medicinal and edible value, and is a plant resource for potential antitumor-led compounds. The chemical constituents of *G. xanthochymus* vary in different plant tissues. Previous review reports documented a number of xanthones that were isolated from barks, stem, twigs, and wood; the flavonoids were mainly extracted from leaves; and the fruit was found to be high in benzophenones and biflavonoids [15,18]. Those reported metabolites have a similar trend to those we tentatively identified via UPLC-QTOF-MS, except fruit, in which few benzophenones were detected (Appendix A). We also presented the comparative analysis of the main metabolites, whose distribution varied in different parts of *G. xanthochymus*. The metabolite profiling of different plant parts will be useful for researchers involved in isolating their metabolites of interest.

The polycyclic polyprenylated acylphloroglucinols (PPAPs) have limited distribution in the Clusiaceae family but many of them have cytotoxic or antitumor effects on human cancer cell lines. Previous studies showed that the polyprenylated benzophenones (belonging to PPAPs) isolated from *G. xanthochymus* fruits displayed potent cytotoxicity in different human cancer cell types [32,33,34]. Recently, 9 new and 12 known PPAPs were isolated from *G. xanthochymus* fruits, and 4 PPAPs showed moderate inhibitory activities against 3 hepatocellular carcinoma cell lines with IC_50_ values in the range of 3.9 to 17.3 μM [35]. However, the fruit extract of *G. xanthochymus* in our study had no significant cytotoxic effect on five cancer cell lines. This may be related to the collected area or harvest time of the fruit, resulting in low contents of PPAPs.

The isolated polyisopentenyl phloroglucinols (**1**–**3**) also belong to a type of PPAPs. The chemical structures of compounds **1** and **2** were similar, but their cytotoxic activities were different. Both of them had the same parent nucleus, and compound **2** presented an extra methylene in C30 that significantly inhibited the growth of cancer cell lines. Structure–activity relationship studies reported that the benzoyl group and prenyl groups in the phloroglucinol skeleton play a crucial role in their cytotoxic effects upon cancer cell lines [36,37,38]. The moiety connected to C-6 of compounds **1**–**2** is important for the cytotoxic activity. In additional, variations in drug response are commonly observed among tumors of the same tissue type in vitro and in vivo [39,40]. Our study showed five different cancer cells responded differently to treatment than compound **2**, but nevertheless positively. Similar results were found with diterpenoids from *Isodon pharicus*, which demonstrated significant inhibitory effects against five human cancer cell lines, with different IC_50_ values ranging from 0.37 to 5.61 μM [41].

The chemical constituents and pharmacological activity of *G. xanthochymus* seed have been investigated less. In the previous research, it was found that the seed mainly contains saturated (34.17%) and unsaturated fatty acids (65.79%) [42]. The seed extract demonstrated antimicrobial activity against *Escherichia coli*, *Bacillus subtilis*, and *Staphylococcus aureus* [43] and strong antidiabetic activity [44]. In this study, we found the seed extract of *G. xanthochymus* demonstrated significant activity against cancer cell lines for the first time. An LC-MS based metabolomics approach was used to obtain 12 potential markers, which were tentatively identified as polyisopentenyl phloroglucinols in the seed extract. Much effort on the mechanism of bioactivity is needed to verify the effectiveness of the *G. xanthochymus* seed extract and polyisopentenyl phloroglucinols. Due to the complexity of the composition of medicinal plants and the limitations of existing separation methods, metabolomics technology combined with the analysis method of bioactivity screening will effectively improve and screen the active ingredients in medicinal plants, and make up for the shortcomings of traditional methods in bioactive-guided isolation; the combined method is an efficient research strategy to obtain the active ingredients of medicinal plants.

## 5. Conclusions

In the present study, we demonstrated an available approach to identify the cytotoxic activities of natural products from *G. xanthochymus* by combining cytotoxicity assays with a UPLC-QTOF-MS-based metabolomics method. A total of 47 metabolites were detected and tentatively identified, which varied in different plant parts. We found that the seed extract had significant cytotoxic effects on five human cancer cell lines (HL-60, A549, SMMC-7721, MDA-MB-231, and SW480) for the first time. Metabolomics-based analysis (PCA and OPLS-DA) of different parts’ metabolites resulted in finding 12 marker compounds from the bioactive seed extract. LC-MS guided the isolation of markers to obtain three polycyclic isopentenyl phloroglucinols. Compound **2** had moderate cytotoxic activity against five human cancer cell types, with IC_50_ values of 14.71~24.43 μM. The results support the potential cytotoxic effects of *G. xanthochymus* and provide a scientific basis for the development and utilization of this plant’s resources.

## Figures and Tables

**Figure 1 metabolites-13-00258-f001:**
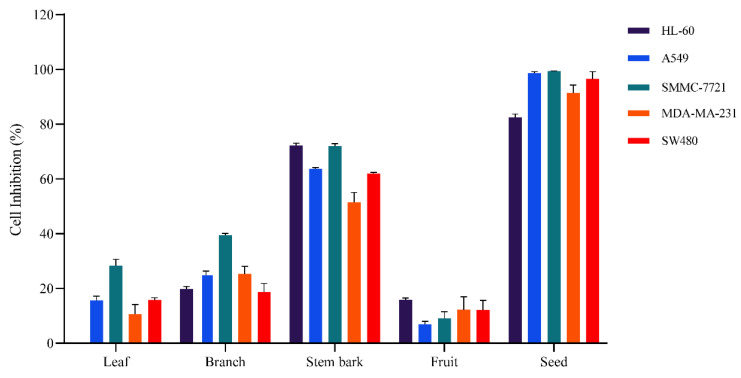
Cytotoxicity assay of different parts of *G. xanthochymus* against five human cancer cells (HL-60, A549, SMMC-7721, MCF-7, and SW480).

**Figure 2 metabolites-13-00258-f002:**
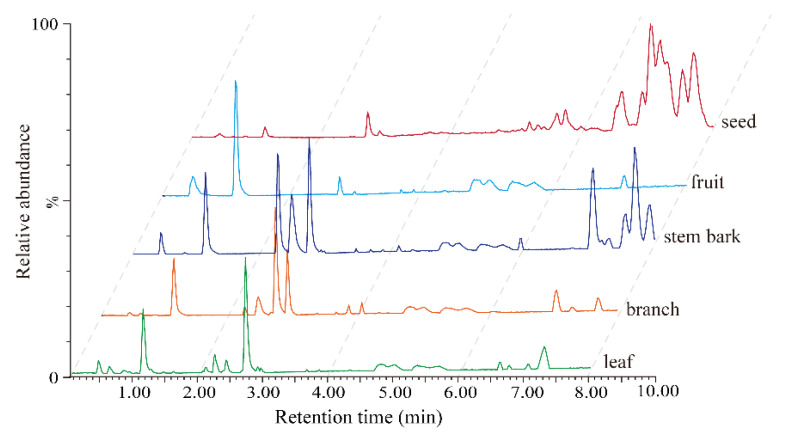
TIC chromatography (negative mode) of different part extracts of *G. xanthochymus*.

**Figure 3 metabolites-13-00258-f003:**
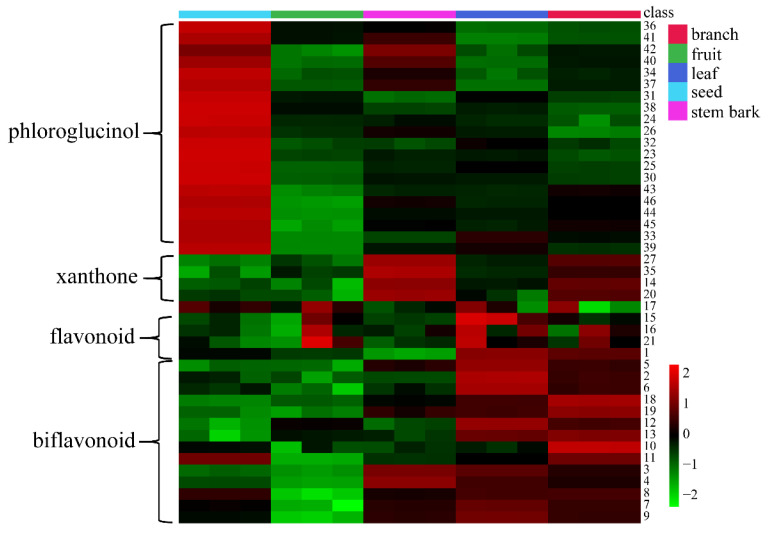
The heatmap presenting the distribution of metabolic profiling in the branch, fruit, leaf, seed, and stem bark extracts of *G. xanthochymus*. The number represents the identified metabolites in Appendix A. Red color indicates a relatively high content. Blue color indicates low content of metabolites.

**Figure 4 metabolites-13-00258-f004:**
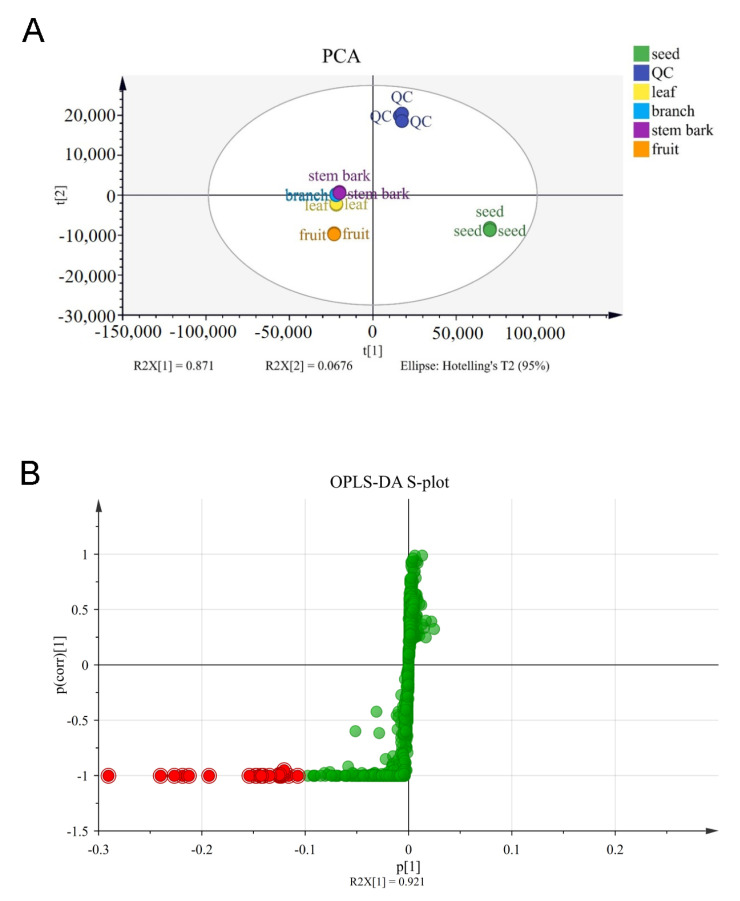
Multivariate analysis of different parts of *G. xanthochymus*. (**A**) Score plot from PCA (negative). (**B**) S-plot from OPLS-DA (negative). Compounds putatively identified in Table 2 are highlighted.

**Figure 5 metabolites-13-00258-f005:**
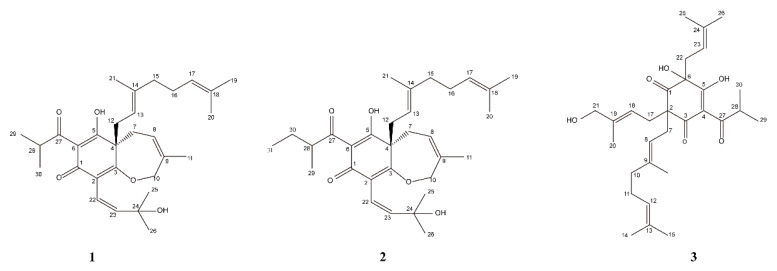
The chemical structure of new compounds **1**–**3**.

**Figure 6 metabolites-13-00258-f006:**
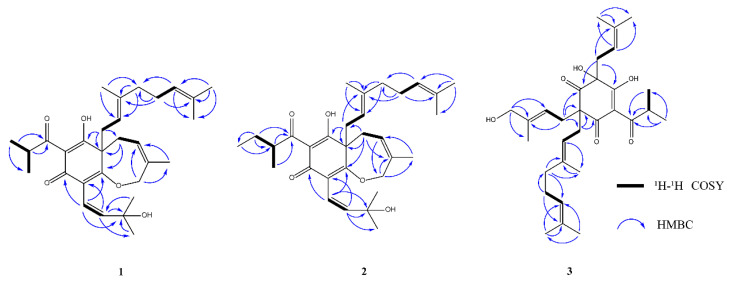
The key ^1^H-^1^H COSY and HMBC correlations of **1**–**3**.

**Figure 7 metabolites-13-00258-f007:**
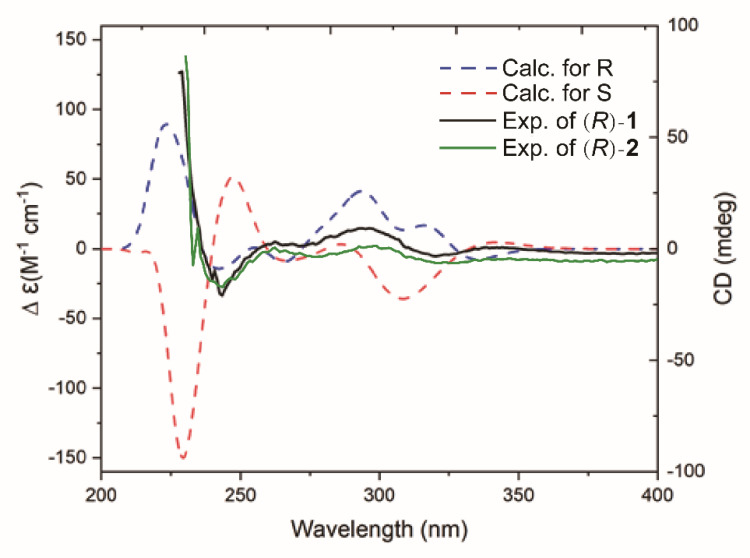
Experimental CD spectrum of **1**–**2** and simulated CD spectra corresponding to the *R* (blue curve) and *S* (red curve) enantiomers of **1**–**2** as calculated.

**Figure 8 metabolites-13-00258-f008:**
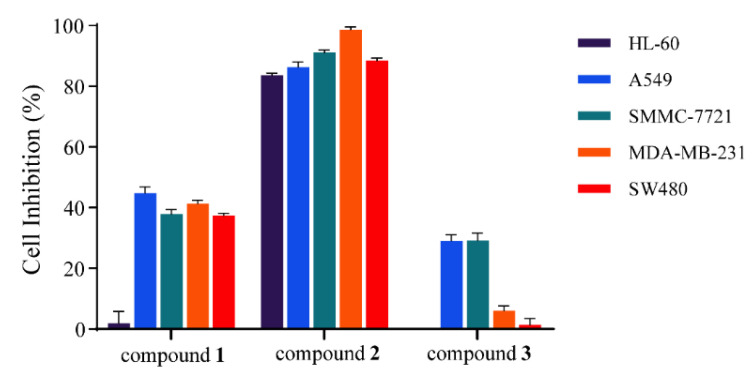
Cytotoxicity assay of compounds **1**–**3** against five human cancer cell lines (HL-60, A549, SMMC-7721, MCF-7, and SW480) at concentrations of 40 μM. Results are expressed as mean ± SD (*n* = 3).

**Table 1 metabolites-13-00258-t001:** Cytotoxic effects of seed extract in IC_50_ (μg/mL) *^a^*.

Samples	Human Cancer Cell Lines				
	HL-60	A459	SMMC-7721	MCF-7	SW480
Seed extract	8.637 ± 0.124	3.436 ± 0.169	2.666 ± 0.130	8.362 ± 0.562	9.438 ± 0.290
DDP *^b^*	1.312 ± 0.024	12.20 ± 0.22	2.527 ± 0.145	18.75 ± 0.55	15.50 ± 0.99
Taxol *^b^*	<0.008	<0.008	0.138 ± 0.008	<0.008	<0.008

*^a^* Values are expressed as the mean ± SD (*n* = 3). *^b^* DDP (cisplatin), Taxol: positive control for cytotoxicity assay.

**Table 2 metabolites-13-00258-t002:** Tentative identification of markers from bioactive fraction parts in seed extract.

Marker	RT (min)	Molecular Ions [M+H]^+^/[M-H]^−^			Molecular Formula	Identification
		m/z	M.F.	ppm		
s1	5.613	517.3531	C_31_H_49_O_6_	0.4	C_31_H_48_O_6_	garcihombronane L [24]
		515.3381	C_31_H_47_O_6_	1.6		
s2	6.595	487.3430	C_30_H_47_O_5_	1.4	C_30_H_46_O_5_	garciosaterpene E or its isomers [25]
		485.3279	C_30_H_45_O_5_	2.5		
s3	6.692	515.3376	C_31_H_47_O_6_	0.6	C_31_H_46_O_6_	unknown
		513.3221	C_31_H_45_O_6_	1.0		
s4	6.944	485.3291	C_30_H_45_O_5_	4.9	C_30_H_44_O_5_	garcinielliptone L or garsubelline A [26,27]
		483.3117	C_30_H_43_O_5_	1.4		
s5	6.996	485.3287	C_30_H_45_O_5_	4.1	C_30_H_45_O_5_	garcinielliptone F [28]
		483.3126	C_30_H_43_O_5_	3.3		
s6	7.208	501.3616	C_31_H_49_O_5_	7.2	C_31_H_48_O_5_	unknown
		499.3446	C_31_H_47_O_5_	4.6		
s7	7.495	499.3446	C_31_H_47_O_5_	4.6	C_31_H_46_O_5_	garsubellin B [29]
		497.3280	C_31_H_45_O_5_	2.6		
s8	7.661	499.3456	C_31_H_47_O_5_	6.6	C_31_H_46_O_5_	garcinielliptone T [24]
		497.3287	C_31_H_45_O_5_	4.0		
s9	8.310	485.3267	C_30_H_45_O_5_	0.0	C_30_H_44_O_5_	garcinielliptone L or garsubelline A [30]
		483.3115	C_30_H_43_O_5_	1.0		
s10	8.907	499.3428	C_31_H_47_O_5_	1.0	C_31_H_46_O_5_	garsubellin E [29]
		497.3276	C_31_H_45_O_5_	1.8		
s11	8.023	483.3117	C_30_H_43_O_5_	1.4	C_30_H_42_O_5_	unknown
		481.2955	C_30_H_41_O_5_	0.2		
s12	8.660	497.3270	C_31_H_45_O_5_	0.6	C_31_H_44_O_5_	unknown
		495.3113	C_31_H_43_O_5_	0.6		

**Table 3 metabolites-13-00258-t003:** ^1^H and ^13^C NMR data (CDCl_3_) of compound **1**–**3**.

Position	1		2		3	
	δ_H_ (*J* in Hz)	δ_C_	δ_H_ (*J* in Hz)	δ_C_	δ_H_ (*J* in Hz)	δ_C_
1		186.9		187.0		205.7
2		106.8		106.9		62.8
3		171.4		171.4		198.0
4		57.8		57.8		111.2
5		196.4		196.5		192.7
6		107.3		107.9		85.1
7	2.50, d (7.1) 2.92, dd (13.5, 10.4)	35.7	2.50, d (6.8) 2.92, ddd (13.5, 10.4, 7.9)	35.9	2.76, dd (13.8, 6.3) 2.86, m	33.9
8	4.90, dd (10.7, 6.0)	122.1	4.90, dd (9.8, 4.2)	122.0	5.05, t (7.4)	118.4
9		138.5		138.6		140.6
10	3.78, d (12.0)	61.8	3.77, m	61.8	1.90, m	40.2
	4.29, d (12.0)		4.30, dd (14.7, 12.2)			
11	1.66, s	22.3	1.65, d (1.2)	22.3	1.97, dd (14.9, 7.0)	26.6
12	2.61, dd (13.7, 7.8) 2.48, d (7.1)	39.1	2.61, dd (13.7, 7.7) 2.48, d (7.2)	39.0	4.97, m	124.0
13	4.81, t (7.2)	117.4	4.80, dd (8.1, 7.0)	117.4		132.0
14		139.5		139.4	1.62, s	26.1
15	1.84, d (6.6)	40.0	1.83, m	40.0	1.52, s	17.8
16	1.87, m	26.9	1.86, m	27.1	1.63, s	16.5
17	4.96, dd (6.7, 5.4)	124.1	4.95, m	124.1	2.89, m 2.27, dd (13.0, 6.3)	40.0
18		131.8		131.8	4.75, m	119.7
19	1.61, s	25.9	1.61, d (0.7)	25.9		141.6
20	1.09, d (6.8)	17.8	1.06, d (6.8)	17.8	1.59, s	21.6
21	1.51, t (3.5)	16.6	1.51, d, (1.9)	16.6	4.07, d (12.2) 3.60, d (12.2)	60.9
22	6.45, d (10.0)	114.8	6.45, d (10.1)	114.8	2.56, dd, (14.7, 9.2) 2.05, dd (13.7, 6.4)	37.2
23	5.36, d (10.1)	123.8	5.35, d (10.1)	123.8	4.85, dd (8.0, 6.6)	115.0
24		81.5		81.5		137.5
25	1.46, s	28.9	1.46, s	28.9	1.43, s	18.2
26	1.45, s	29.0	1.45, s	29.0	1.63, s	25.8
27		208.1		207.6		207.1
28	3.92, m	35.8	3.81, dd (13.7, 6.9)	42.1	3.44, m	35.3
29	1.07, d (6.8)	19.3	1.10, m	16.4	1.27, d (6.8)	20.5
30	1.12, d (6.8)	18.8	1.33, m	26.9	1.12, d (6.7)	18.0
31			0.92, t (7.4)	12.0		

**Table 4 metabolites-13-00258-t004:** Inhibitory effects of **2** on human cancer cell lines in IC_50_ (μM) *^a^*.

Compound	Cancer Cell Lines				
	HL-60	A459	SMMC-7721	MDA-MB-231	SW480
2	19.02 ± 1.26	24.43 ± 0.42	15.69 ± 0.57	16.29 ± 1.44	14.71 ± 0.33
DDP *^b^*	16.07 ± 0.32	28.06 ± 2.29	24.44 ± 0.75	26.69 ± 0.50	28.83 ± 0.83
Taxol *^b^*	<0.008	<0.008	0.312 ± 0.019	<0.008	<0.008

*^a^* Results are expressed as the mean ± SD (*n* = 3). *^b^* DDP (cisplatin), Taxol: positive control for cytotoxicity assays.

## Data Availability

The other data that support the findings of this study are available in the Appendix A of this article.

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
