# Peer review of "Cytotoxic Isopentenyl Phloroglucinol Compounds from Garcinia xanthochymus Using LC-MS-Based Metabolomics"

_metabolites, 2023, doi:10.3390/metabo13020258_

Round 1

Reviewer 1 Report

Several chemical metabolites with significant anti-tumor actives isolated from Garcinia species has been researched in recent years. Authors aimed to identify bioactive compounds from different parts of Garcinia xanthochymus through combining LC-MS-based metabonomcis with anti-tumor assays. They found out that 70% methanol seed extract possessed significant anti-tumor effect on five human tumor cell lines (HL-60, A549, SMMC-7721, MDA-MB-231 and SW480). They discovered  that 12 potential markers from seed extract may relate to bioactive. Among them, garxanthochin B demonstrated significant cytotoxic activity against five human tumor cells with IC50 values of 14.71~24.43 μΜ, which was similar to that of the positive control. Hence, garxanthochin B has potential applications in the development of antitumor lead natural compounds. This manuscript reviews 40 articles and provide a complex survey of current literature dealing with this topic. The topic of this manuscript is up to date, interesting and well suited for  Journal Metabolites. The manuscript is well written and divided into 5 main parts, the text is clear and easy to read. For better understanding authors used 8 illustrations and 4 tables. This aid the readers understanding. I suggest checking for some small spelling mistakes and grammar errors. Otherwise, I have no major concerns about this manuscript and I recommend it for publication.

Author Response

Dear reviewer,

Thank you very much for your comments and professional advice. We have revised the manuscript according to your comments and suggestions. Our revised manuscript has undergone English language editing by MDPI.

Thank you very much for your attention and time. Looking forward to hearing from you.

Your sincerely

Ping Li

South China Agricultural University.

Reviewer 2 Report

The authors explored the antitumor activities of phloroglucinol compounds isolated from Garcinia xanthochymus seed. A biochemometric approach was used to identify potential active compounds which were validated through isolation of the compounds and testing for activity. 

Here are a few concerns

1. Editing is required espeically by a professional 

2. I think it should be "metabolomics" and not "metabonomics"

3. No justification was presented for selecting the different plant part?

4. No justification for selecting the biological activities. Literature proived did not indicate traditional use for treatment of tumor 

5. Line 179 ...it should be "raw data" not "row data"

6. Line 204 should be "most active"

7. LC-MS data for metabolomics was collected using positive and negative modes. Why was the result for positive mode not presented?

8. Line 255... the heatmap indicates intensity and it is displayed in a range from one colour to another. The samples were not divided into two categories as indicated by authors

9. The findings were not adequately discussed. What was the significance of the seed having the most active activity? What was the advantage of the study approach compared to classical bioactive guided approach? What is unique about phloroglucinol. Is there any taxonomical relevance?

Author Response

Dear reviewer,

Thank you very much for your comments and professional advice. We have revised the manuscript according to your comments, suggestions, and remarks. All revisions and corrections are marked using the “Track Changes” in the revised manuscript. Attachment is the details of our responses to your comments.

Thank you very much for your attention and time. Looking forward to hearing from you.

Your sincerely

Ping Li

South China Agricultural University.

Reviewer 3 Report

The article describes the cytotoxic activity of the crude extract of different parts (leaf, branch, stem bark, fruit and seed) of G. xanthochymus and of compounds isolated from the plant against five human cancer cell lines.

The English need to be edited

In relation to the processing of the plant and analytical procedure I have the following comments:

The quality of metabolomic analyzes depends on biomaterial collection and processing protocols, including drying, grinding, and extraction methods, so the risk of artifact formation in the extraction and concentration of solvents used is high. There are many examples of various artifacts formed through chemical reactions between solvents (methanol, ethanol, etc.) or contaminations with functional groups in the analytes. When samples are dried in a rotary evaporator at low pressure and medium temperature, there is a risk and many times compounds are produced that generate false positives, even the generation of molecules that we may believe are new. In the work the case of concentration to dryness in a rotary evaporator is presented and for this reason it is not possible to adequately evaluate all the information presented.

Regarding the biological activity, I make the following suggestions:

The authors must define which of all the activities mentioned corresponds this research: anticancer, antineoplastic, antitumoral or cytotoxic activity, because they are not synonims

Cell activity was performed against five human tumor cell lines (HL-60, A549, SMMC-7721, 136 MCF-7 and SW480): the authors should describe the tissue types corresponding to the cell lines, the vendor codes and justify why those lines were choosen for the study

Quantities in miles like: 15,000 should be set to 15,000 and signs like ~ should be enhanced to scientific standards.

For cytotoxic activity, the authors report concentrations of 3000 to15,000 cells per well, what was the criteria for this difference in cell number? The number or percentage of confluence used in assays needs to be more precise.

In methodology the cell viability assay needs to be rewritten, it is not understood how the treatments were performed on the cells.

The Reed and Muench method needs a reference and the GraphPad Prism software needs a version

Line 206: “significant cytotoxicity against five cells”: explain or adjusted this sentence please

Tables: define DDPb

Figure 8A: indicate the concentration of compounds 1, 2 and 3 used

Figure 8 (B and C): Why do the authors show the cell viability of the positive controls but not that of the tested compounds?

The discussion is very poor, it must be completely adjusted, including again the explanation of the difference found between the analyzed cell lines.

The format of the references in the text must be adjusted.

Author Response

(The authors gave the same response as above.)

Reviewer 4 Report

The manuscript entitled “Antitumor Activity of Isopentenyl Phloroglucinol Compounds from Garcinia xanthochymus by LC-MS-based Metabolomics Analysisby Li et al. reported the isolation and structural elucidation of three new isopentenyl phloroglucinols (garxanthochin A-C) based on LC-MS metabolomics analysis combined with cytotoxic assays. The authors have found the MeOH extract of Garcinia xanthochymus seeds showed the most significant cytotoxicity. 47 compounds were identified in diferent parts of G. xanthochymus. In addition, 12 potential markers were identified from seed extract using LC-MS-based metabolomics analysis including PCA and OPLS-DA. Compound (2) garxanthochin B displayed significant cytotoxic activity against five human tumor cells with IC50 values of 14.71~24.43 μΜ

In my opinion, the manuscript is quite good. However, the manuscript should be carefully revised before publication.

1.      What does “metabonomics” mean? Is it “metabolomics” ? The authors used this word “metabonomics” in the title of paper and many times in the manuscript. I have highlighted

2.      Please check many English errors. I have highlighted in the manuscript such as:

Line 126, 129: Sephadex

Line 162: acetonitrile

Line 13, line 33, line 428: Rephrased the sentences

Line 196-208: Many sentences were meaning duplicated

Line 291: remove “is”

3.      I could not find where the reference 1 is cited in the manuscript. Please correct the line 34 (Sung et al., 2021)

4.      The reference 40 is not needed.

5.      Line 313, 341: 10 degrees of unsaturation

6.      Line 375: 9 degrees of unsaturation

Line 318, 350, 386: two isopentenyl units can be replaced by geranyl group.

Line 400: “Compound 2”, the character “C” is normal.

 Check the HRMS data of compound 3 in the SI, it’s not matched with the one in the manuscript

Author Response

(The authors gave the same response as above.)

Round 2

Reviewer 2 Report

Authors have effected all needed changed

Author Response

Dear reviewer,

Thank you very much for your comments and suggestions. 

best regards

Ping Li

Reviewer 3 Report

The authors have made important changes to improve the quality of the article however:

1. Previously, a comment related to the processing of the plant was made, it was indicated that according to the procedure carried out, there was a risk of artifact formation during the extraction and concentration of solvents, so there is no certainty of the quality of the products formed and presented here. Authors must ensure and demonstrate that no artifacts or compounds occurred in this study that would lead to false positives.

2. The title in supplementary material should also be changed

Author Response

Dear reviewer,

Thank you very much for your comments and professional advice. Our responses to your comments as below:  

Point 1:  Previously, a comment related to the processing of the plant was made, it was indicated that according to the procedure carried out, there was a risk of artifact formation during the extraction and concentration of solvents, so there is no certainty of the quality of the products formed and presented here. Authors must ensure and demonstrate that no artifacts or compounds occurred in this study that would lead to false positives. 

Response 1: Thank you for your comments. Firstly, for the preliminary screening analysis of activity, 70% methanol was used to extract different parts of Garcinia xanthochymus. The filtered solution was concentrated using rotary evaporator under reduced pressure (0.098Mpa) at 40 °C to yield a crude extract. In the process of concentration, the boiling point of methanol is lower than that of water in 70% methanol-water solution, and methanol is first evaporated, followed by water. Therefore, under the condition of reduced pressure concentration, residual water may have a small amount, while the organic solvent residue is very limited in the extract. In our study, we have dried all the solvent in each evaporation. Secondly, all the compounds were purified using semi-preparative HPLC, eluted with a gradient solvent system of acetonitrile-water. In order to avoid the denaturation of the compounds, the solution of target peaks was collected in HPLC and then used to thoroughly dry by nitrogen, so that we prepared almost no solvent residue of the pure compounds. We have added “All the solution of target peaks in semi-preparative HPLC were thoroughly dried by nitrogen” in the Extraction and Isolation. To sum up, we ensure that the samples or compounds used in the cell viability assay are no solvent residue and there are no false positive results. 

Point 2:  The title in supplementary material should also be changed.

Response 2: The title in supplementary material have been corrected. 

Thank you very much for your attention and time. Looking forward to hearing from you.

Your sincerely

Ping Li

Reviewer 4 Report

The authors have made corrections to improve the manuscript.

However, the authors should correct some errors:

Page 1, line 20: Change “bioactive” to “bioactivity”

Page 2, line 91: methanol

Line 202, 205: Change “70% seed methanol extract” to “70% methanol seed extract”

Line 317: Change “two isopentenyl units (C-12 to C-21)” to a geranyl unit (C-12 to C-21). You can remove the sentence “H-17 and C-15, H-16 and C-15, C-14, H-15 and C-17, C-16 had HMBC correlation, which proves that the geranyl group was con
nected through C-15”.

Table 3. Check the NMR data. Compound 2 data in C-30 and C-31 are not correct.

Line 381: Change “two isopentenyl units” to “a geranyl unit”

Line 457: Isodon pharicus in italic

Author Response

Dear reviewer,

Thank you very much for your comments and professional advice. Our responses to your comments as below:

Point 1: Page 1, line 20: Change “bioactive” to “bioactivity”

Response 1: We have corrected it.

Point 2: Page 2, line 91: methanol

Response 2: We have corrected it.

Point 3: Line 202, 205: Change “70% seed methanol extract” to “70% methanol seed extract”

Response 3: We have corrected it.

Point 4: Line 317: Change “two isopentenyl units (C-12 to C-21)” to a geranyl unit (C-12 to C-21). You can remove the sentence “H-17 and C-15, H-16 and C-15, C-14, H-15 and C-17, C-16 had HMBC correlation, which proves that the geranyl group was connected through C-15”.

Response 4: We agree with your suggestion. We have changed “two isopentenyl units (C-12 to C-21)” to “a geranyl unit (C-12 to C-21)” and removed the sentence “H-17 and C-15, H-16...”

Point 5: Table 3. Check the NMR data. Compound 2 data in C-30 and C-31 are not correct.

Response 5: Thanks for your careful checks. We have double checked the NMR data and corrected the C-30 and C-31 in the Table 3.

Point 6: Line 381: Change “two isopentenyl units” to “a geranyl unit”.

Response 6: We have corrected it.

Point 7: Line 457: Isodon pharicus in italic.

Response 7: We have corrected it.

Thank you very much for your attention and time. Looking forward to hearing from you.

Your sincerely

Ping Li

Round 3

Reviewer 3 Report

Dear authors, thank you for improving this article. I agree with the information described there. I have no more comments